# Analysis of the Influence of Fabrication Errors on the Far-Field Performance of Si and Si$_3$N$_4$ Antennas

Yifan Xin [1,2], Wenyuan Liao [3], Lei Yu [1,2], Pengfei Ma [1,2], Zheng Wang [1,2], Yibo Yang [1,2], Licheng Chen [1,2], Pengfei Wang [1,2,\*], Yejin Zhang [1,2] and Jiaoqing Pan [1,2,\*]

[1] Key Laboratory of Optoelectronic Materials and Devices, Institute of Semiconductors, Chinese Academy of Sciences, Beijing 100083, China; xinyifan@semi.ac.cn (Y.X.); yulei@semi.ac.cn (L.Y.); pfma@semi.ac.cn (P.M.); wangzheng@semi.ac.cn (Z.W.); yangyibo@semi.ac.cn (Y.Y.); lichengchen@semi.ac.cn (L.C.); yjzhang@semi.ac.cn (Y.Z.)

[2] Center of Materials Science and Optoelectronics Engineering, University of Chinese Academy of Sciences, Beijing 100049, China

[3] Science and Technology on Reliability Physics and Application Technology of Electronic Component Laboratory, Guangzhou 511370, China; wyliaophd@126.com

\* Correspondence: pfwang15@semi.ac.cn (P.W.); jqpan@semi.ac.cn (J.P.)

**Abstract:** Antennas are important components in optical phased arrays. However, their far-field performance deteriorates when random phase noise is introduced because of fabricating errors. For the first time, we use a finite-difference time-domain solution to quantitatively analyze the far-field characteristics of Si and Si$_3$N$_4$ antennas considering process errors. Under rough surface conditions based on a fishbone structure, we find that the quality of the main lobe of the Si antenna deteriorates badly, with −0.87 dB and −0.51 dB decreases in the sidelobe level and 5.78% and 3.74% deteriorations in the main peak power in the φ (phase-controlled) and θ (wavelength-controlled) directions, respectively. However, the Si$_3$N$_4$ antenna is only slightly impacted, with mere 0.39% and 0.71% deteriorations in the main peak power in the φ and θ directions, respectively, which is statistically about 1/15 of the Si antenna in the φ direction and 1/5 in the θ direction. The decreases in the sidelobe level are also slight, at about −0.08 dB and −0.01 dB, respectively. Furthermore, the advantages of the Si$_3$N$_4$ antenna become more remarkable with the introduction of random errors into the waveguide width and thickness. This work is of great significance for the design and optimization of OPA chips.

**Keywords:** LiDAR; optical phased array; optical antenna design fabrication error; silicon; silicon nitride

## 1. Introduction

Optical phased array (OPA)-based Light Detection and Ranging (LiDAR) is a system that actualizes two-dimensional beam steering using phase and wavelength control, achieving the same functionality as traditional radars on a chip [1–6]. Due to its high anti-interference ability and response speed, it has broad application prospects in fields such as meteorology and the emerging prospect of autonomous driving, which has recently received significant attention [7–11]. Since the concept of OPAs was first proposed, it has developed rapidly [12–15]. Researchers nowadays are dedicated to designing and manufacturing OPA structures with several essential parameters, including a larger array scale, larger field of view and greater sidelobe level (SLL), which are all important for practical usage. Up to now, integrated OPAs with even 1 × 8192 channels have been designed and fabricated [16]. The field of view represents the maximum steering angle and is limited by the far-field grating lobe. The performance begins to suffer when it reaches a certain angle. There are several specially designed structures for weakening or bypassing the impact of the grating lobe, such as irregularly or aperiodically distributing antennas using an optimization algorithm [17,18] and splicing the field of view using optical switching [19,20].

The irregular antenna structures break the interference condition and disperse energy into bottom noise. This method can lift the restriction of the grating lobes, allow for a relatively large turning angle and cover a fairly large field in the horizontal direction. A $140° \times 19.23°$ field of view has been realized, but it causes an inevitable increase in the SLL. The splicing field array bypasses the influence of the grating lobe and splices every single field of view into a large one, but the complexity of the system inevitably increases. As for realistic usage, such as in automated driving, the steering angle in the vertical direction is also essential. The field range required is generally believed to be greater than $30°$ [21]. Steering in the θ direction is usually realized using wavelength tuning. Other methods have been put forward recently, such as a polarization multiplexing OPA [22–24]. A polarization switch accompanied by superlattice grating antennas was designed, using both the TE mode and TM mode and doubling the FOV to $24.8° \times 60°$ [24]. The SLL is important when it comes to the actual usage of detection and measuring. The receiving system, which is designed for the reflection of the main peak power, may be cheated using the reflection of the sidelobe power if the sidelobe is not slight enough. The SLL is generally below $-10$ dB.

Various OPA structures have been proposed to achieve a larger FOV and greater SLL. In general, OPAs are usually composed of input waveguides, phase shifters and antenna arrays. Gratings are etched onto waveguides to radiate power, and the radiations from the antenna arrays interfere and form the far-field pattern for further detection and measurement. The sidewall grating antenna is widely used with various types of phase shifter structures [25,26]. Equally, dual-layer grating antennas are also designed and manufactured for a larger aperture in the wavelength-tuning direction. These have an additional $SiO_2$ layer between the Si waveguide and gratings [27,28]. Additionally, more complex structures, such as $Si_3N_4$ perturbation grating structures on the Si waveguide [29–31] and, furthermore, 2D fishbone surface grating structures, have been put forward. A compressive divergence angle of $0.02°$ and large aperture have been reported [32]. Another type of antenna was reported and manufactured recently with a high-contrast grating (HCG) structure [33,34], which etches gratings onto an integrated slab including a low-refractive-index interlayer and a high-refractive-index grating layer. This structure improves the upward radiation efficiency. It also bypasses the large pitch between the antenna channels and increases the scanning range.

Nowadays, OPA arrays are usually fabricated on 8-inch silicon-on-insulation (SOI) wafers. Although the structures designed vary, the OPA arrays usually occupy a small area of about several square millimeters, so they are convenient for integrated manufacturing. To fabricate antenna arrays, chemical polishing and dry etching processes are usually used. The polishing process ensures that the smoothness of the surface parallels that of the epitaxial layers. And the etching process achieves different grating structures along the waveguides. However, in the design of most OPA chips, the impact of fabrication errors has generally been left out. This introduces additional phase noise into the antenna array. The phase error of the antenna front end can be calibrated using a phase shifter, but the phase noise inside the antenna cannot be eliminated. And it is fatal to the far-field performance. Fabrication errors lead to a significant gap between the designed result and actual performance in the far field. The obvious increase in bottom noise is the most notable feature observed. Therefore, it is necessary to comprehensively analyze the impact of fabrication errors on antennas.

In this paper, we quantitatively analyze the impact of fabrication errors on the far-field performance of antennas using a finite-difference time-domain (FDTD) solution for the first time. Considering the two most important fabrication processes, we construct and simulate a rough surface based on Gaussian distribution and random errors in the widths of the antenna arrays, both of which take the etching process into account. And for the polishing process, random errors in the thicknesses of the antenna arrays are discussed as well. To better explain the reason behind the influence of fabrication errors, we made comparisons between a Si antenna and $Si_3N_4$ antenna under identical system settings and using the same structure type by analyzing the impact of the intensity distribution in the

far field. The results show that the fabrication error has a more significant impact on the Si antenna, including the loss of main peak intensity and the distortion of spot and larger bottom noise. Compared with that observed in the Si antenna, the influence on the $Si_3N_4$ antenna's performance under the same distribution of a rough surface is much smaller than that exerted on the Si antenna. Considering both the sidelobe level (SLL) and the quality of the main peak, the variation in the main peak power is only 1/15 of the Si antenna in the $\varphi$ direction and 1/5 in the $\theta$ direction. As far as we know, this is the first time that the influence of process errors on the far field of antennas has been analyzed quantitatively, and it has guiding significance for the design of OPAs.

## 2. Simulation and Analysis

Si is a material traditionally used in OPAs because of its high refractive index and well-established technology, which allows for compact designs. Compared with silicon antennas [23,35], silicon nitride ($Si_3N_4$) antennas have become increasingly popular in recent years due to their broader transparent working band, lower refractive index contrast and reduced third-order nonlinear effects [36,37]. Both types of antennas are compatible with CMOS technology. Based on a series of previous experimental results [31], it was found that an OPA with a Si antenna had higher background noise than expected, with a more diffuse energy distribution. In addition, the energy along the $\varphi$ direction was no longer symmetrical, and the brightness of the grating lobes and sidelobes consistently exceeded expectations. The impact reduced the power of the main lobe and disturbed the measurement, as shown in Figure 1. Figure 1a displays the far-field pattern of an OPA with a Si sidewall-etched grating antenna array on a 220 nm SOI substrate with 128 channels. The pitch between adjacent antennas was 2 μm. The far-field pattern was calibrated, since the OPA with a Si antenna required optimization before actual measurement, as it exhibited poor performance. The waveguide width was 0.5 μm to enable single-mode propagation. The inward etching depth was 0.05 μm, and the grating period was 0.72 μm. Figure 1b displays the far-field pattern of a $Si_3N_4$ sidewall-etched grating antenna array, which supports a single mode with a width of 1.5 μm, an inward etch of 0.15 μm, and a thickness of 200 nm. Figure 1 also presents detailed parameters of the two materials. The performance of the Si antennas was notably inferior to that of the $Si_3N_4$ antennas, even without special calibration. Taking into consideration the actual process steps, we first conducted a quantitative analysis on antenna arrays with a rough surface caused by the etching process to explore the factors that influence antenna performance.

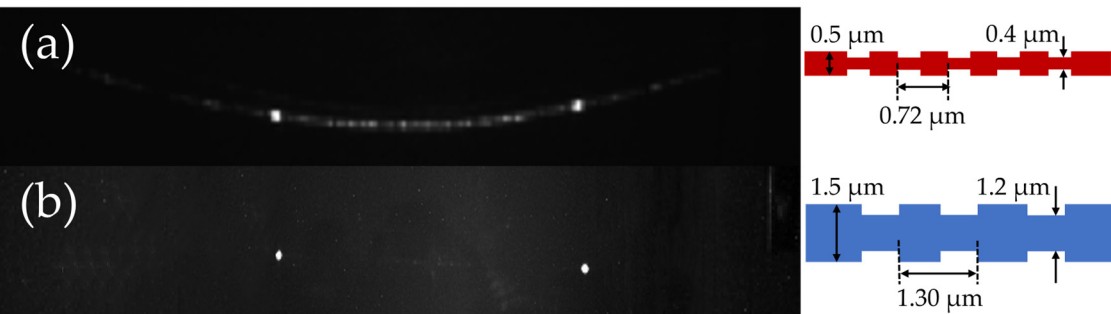

**Figure 1.** Far-field experimental results and detailed parameters of (**a**) Si and (**b**) $Si_3N_4$ antennas.

We used a finite-difference time-domain (FDTD) solution to conduct our simulations. The input sources had wavelengths of 1550 nm (TE), and the perfectly matched layer (PML) boundaries were configured to absorb all radiation to the boundaries and avoid reflection. According to previously fabricated antennas, we based our design on the parameters mentioned above and created a basic antenna array with four channels spaced 4 μm apart. This design aims to prevent crosstalk and to align with the $Si_3N_4$ array that will be compared below (shown on the left side of Figure 2). The duty cycle was 0.5. The

expected symmetric far-field pattern was obtained (as shown in Figure 3 with blue lines). To investigate the effect of rough burrs on the far-field quality of the antenna, a rough surface morphology was introduced to the smooth antenna. Dry etching is commonly utilized in the fabrication of gratings, where plasma is directed perpendicular to the substrate to bombard the waveguide. Given the consideration that the thicknesses of the waveguides were small for both the Si and Si$_3$N$_4$ antennas (220 nm and 200 nm), we disregarded the inclination of the plasma beam and developed the model with a 2D (x and y directions) rough surface, as shown in Figure 2.

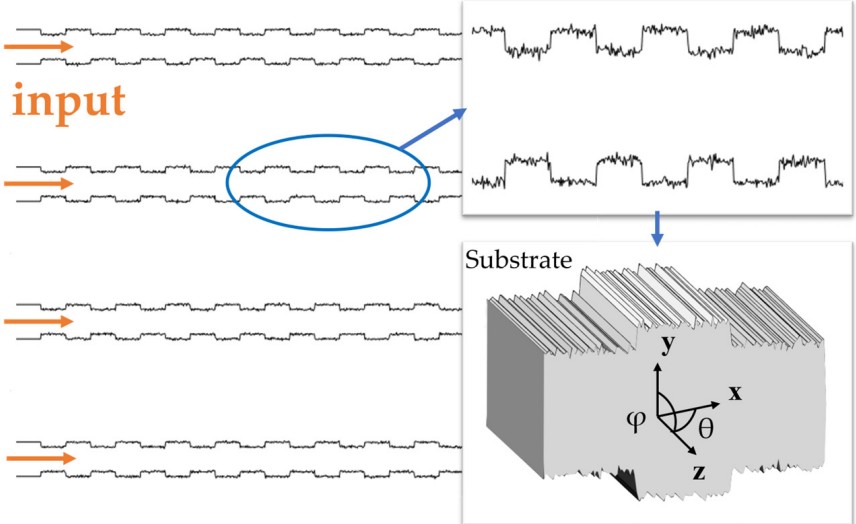

**Figure 2.** Sketch map of the rough burr surface on both sides of the antenna (array).

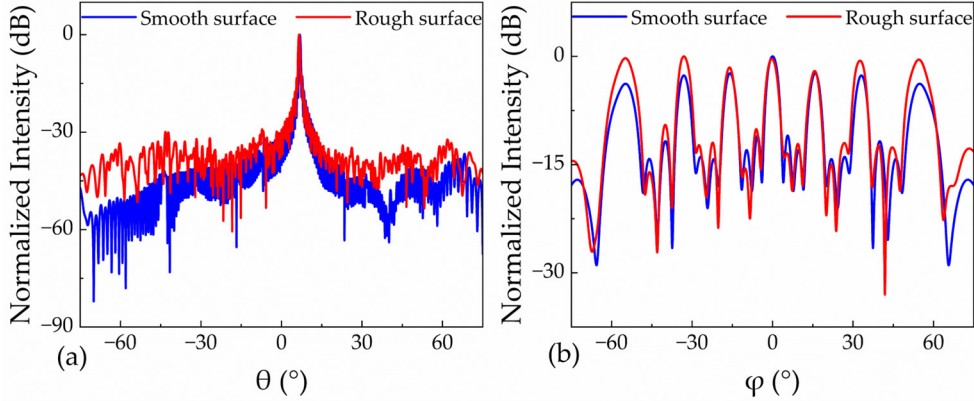

**Figure 3.** Far-field distribution of Si antennas with smooth and rough surface morphologies is shown in (**a**) the θ direction and (**b**) the φ direction.

The size of the roughness burrs conformed to a Gaussian distribution [38,39], with a standard deviation ($\sigma$) of 10 nm, a mathematical expectation ($\mu$) of 0 and a correlation length of 20 nm. The probability density function is expressed as

$$f(x) = \frac{1}{\sqrt{2\pi}\sigma} \exp\left(-\frac{(x-\mu)^2}{2\sigma^2}\right)$$

where $x$ represents the distance between the peak of a rough burr and the original smooth surface. High-precision meshes are set around the antenna cells, taking into account the correlation length and the limitations of simulation memory caused by the detailed structure of rough surfaces. Figure 3 displays the far-field results of Si antennas with smooth and rough surfaces with the same level of accuracy.

To assess the quality of the far-field light spot, we utilized the proportion $\eta$ of the main peak energy (referred to as $E_m$) in the upward radiation (referred to as $E$) as an indicator of the main lobe's quality.

$$\eta = \frac{E_m(3dB)}{E}$$

A larger $\eta$ indicates more concentrated energy and a higher quality of the main peak, while a lower $\eta$ results in higher background noise. The background noise in the θ direction (as indicated in Figure 2) of the smooth model was concentrated between −40 and −50 dB. The SLL was measured at −12.81 dB. The $\eta$ was calculated to be 70.37%. In the φ direction, the far-field energy distribution of the smooth model was symmetrical, as shown by the blue line in Figure 3b. The sidelobe level (SLL) was −10.97 dB, and the $\eta$ was 15.02%. Additionally, the grating lobe level was 2.34 dB lower than the main peak.

After the rough surface was added, the noise level increased significantly. Because the rough surface was randomly built, three sets of simulated random surface structures were used to ensure more convincing conclusions. One of the far-field results is depicted in red lines in Figure 3, while detailed results for both the SLL and $\eta$ in both directions are presented in Table 1 below. In the θ direction, the background noise increased significantly across the entire field of view by approximately 10 dB (refer to Figure 3a). This effect was more pronounced after calculating $\eta$, with decreases from 70.37% to 66.76%, 6.95% and 66.19%, respectively. Additionally, the SLL deteriorated to −12.00 dB, −12.28 dB and −12.63 dB, respectively. The average SLL was −12.30 dB, with a decrease of 0.51 dB compared with the smooth model. The average $\eta$ was 66.63%, which is 3.74% lower than that of the smooth model. In the φ direction, the energy distribution was no longer symmetric, leading to a significant deterioration of both the SLL and the quality of the main lobe (see Figure 3b). Upon further calculation, it was found that $\eta$ decreased to only 9.38%, 9.14% and 9.19%, respectively. The average $\eta$ was 9.24%, with a 5.78% decrease compared with the smooth model. The SLL values were −9.74 dB, −10.88 dB and −9.68 dB, respectively. The average SLL was −10.10 dB, with a decrease of 0.87 dB compared with the smooth model. The peak energy of the grating lobe also increased, exceeding that of the main peak. These indicate a noticeable rise in background noise. It was proven that the background noise evidently rose. Additionally, it is possible that the actual fabrication error is more complex and operates on a larger scale. For example, the $\mu$ in a Gaussian distribution may deviate from 0, and the phase noise may be more pronounced. The structure will be further discussed below.

**Table 1.** Comparison of parameters when using different materials.

| Materials | φ Direction | | | | θ Direction | | | |
| | SLL | | η | | SLL | | η | |
| | Smooth | Rough | Smooth | Rough | Smooth | Rough | Smooth | Rough |
| --- | --- | --- | --- | --- | --- | --- | --- | --- |
| Si | −10.97 dB | −9.74 dB | 15.02% | 9.38% (−5.64%) | −12.81 dB | −12.00 dB | 70.37% | 66.76% (−3.61%) |
| | | −10.88 dB | | 9.14% (−5.88%) | | −12.28 dB | | 66.95% (−3.42%) |
| | | −9.68 dB | | 9.19% (−5.83%) | | −12.63 dB | | 66.19% (−4.18%) |
| | | | Average: 9.24% (−5.78%) | | | | Average: 66.63% (−3.74%) | |
| Si$_3$N$_4$ | −12.34 dB | −12.26 dB | 18.55% | 18.26% (−0.29%) | −13.00 dB | −13.01 dB | 69.24% | 68.27% (−0.97%) |
| | | −12.29 dB | | 18.10% (−0.45%) | | −12.98 dB | | 68.69% (−0.55%) |
| | | −12.23 dB | | 18.12% (−0.43%) | | −13.00 dB | | 68.64% (−0.60%) |
| | | | Average: 18.16% (−0.39%) | | | | Average: 68.53% (−0.71%) | |

The difference in the far field between the smooth and rough models is attributed to the phase noise caused by the burrs. When light beams pass through the rough burrs between the antenna grating (with an index of 3.45) and the $SiO_2$ cap (with an index of 1.45) and are radiated into the surroundings, they experience an optical path difference in different channels. This leads to phase noise when interference occurs, resulting in an unexpected far-field distribution and a deterioration in the quality of the main peak.

To compare the performance of two materials under the same fabrication process, the $Si_3N_4$ model was augmented with a Gaussian distribution rough surface morphology. To achieve single-mode transmission, the $Si_3N_4$ antenna was set with a thickness of 200 nm and a width of 1.5 μm. These parameters are commonly used in practical applications. The same ratio of inward etching depth, about 0.15 μm, was introduced into the antenna for comparison with the Si antenna. The far-field results of both smooth and rough surface antenna arrays are shown in Figure 4. It can be observed that the rough surface morphology had minimal impact on the background noise in the θ direction across the entire FOV (Figure 4a). After calculation, the η decreased slightly from 69.24% to 68.27%, 68.69% and 68.64%, respectively, representing an average decrease of 0.71%. In the φ direction, the proportion of the main peak energy in the overall upward radiation changed from 18.55% to 18.26%, 18.10% and 18.12%, with an average decrease of only 0.39% according to Figure 4b. This change is relatively insignificant compared with that observed in the Si antenna arrays simulated previously. And it is only approximately 1/15 of the Si antennas in the φ direction and 1/5 in the θ direction, with the same antenna spacing, length and channel number. According to the results, the surface roughness has no significant effect on the SLL (a deterioration of 0.08 dB in the φ direction and 0.01 dB in the θ direction on average) or on the level of the grating peaks. The detailed results of each random simulation are also shown in Table 1. For each random model, the difference between the $Si_3N_4$ model and the Si model exceeded 1/12 and 1/3, respectively, in two directions. The results of all three random simulations remain within relatively stable ranges, which makes the data and conclusions convincing.

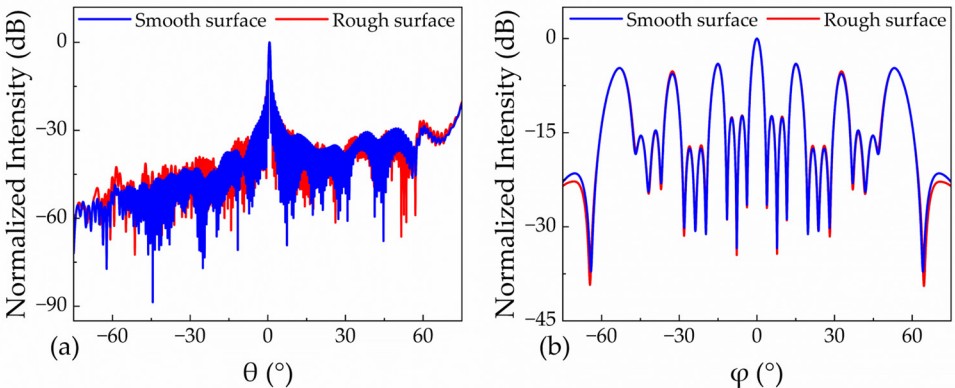

**Figure 4.** Far-field distribution of $Si_3N_4$ antennas with smooth and rough surface morphologies (**a**) in the θ direction; (**b**) in the φ direction.

The different behavior of the two antennas on rough surfaces was due to the difference in the refractive indices of the materials. Si has a higher refractive index of about 3.45 compared with $Si_3N_4$, which has a refractive index of about 1.99. As a result, the Si waveguide has a greater capacity to confine light, leading to a narrower width and a relatively larger proportion of rough surface area. The optical field is more affected by dimensional errors induced during the fabrication process. Furthermore, the difference in refractive index also led to a larger phase error than that of the $Si_3N_4$ antenna when light went through the rough burrs. To summarize, the manufacturing error had a greater impact on the Si antenna than on the $Si_3N_4$ antenna.

The effective length is an important parameter for antennas. Different effective lengths lead to different effective apertures, which represent the efficiency of receiving and trans-

mitting OPAs. This serves as a guide for fabrication, as longer antennas may not necessarily be more efficient. They may result in faint power remaining in the antenna waveguides and faint radiation being produced. To better benefit the actual design of antennas, Si antennas with multiple effective lengths are simulated to explore the trends of the influence of the rough surface on the variation in the effective length. The effective length is defined as the length at which the input power decays to $1/e^2$. To ensure the persuasiveness of our findings, we conducted simulations for three sets of results for each set of antenna parameters on a randomly built rough surface. Table 2 displays the detailed results, including the corresponding data and average values. It is evident that the background noise of the rough-surfaced Si antenna increased gradually in both directions as the effective length increased. Simultaneously, the energy proportion of the mean peak decreased, especially in the θ direction. This decrease ranged from −2.68% in the 50 μm array to −4.62% in the 130 μm array. The SLL deteriorated more in the φ direction compared with the smooth model and showed a positive correlation with an effective length from a statistical perspective. However, the fluctuation range was relatively large for all three sets of antenna parameters due to the random phase noise that added to each angle randomly. The results indicate that the impact of rough surfaces on the background noise is more significant when the antenna array has a longer effective length. The longer the effective length of the antenna, the shallower the etching depth, which amplifies the influence of rough surface disturbance on the etching groove. Regarding $Si_3N_4$ antennas' long effective length (~mm), due to the rough surface and limited computer capacity, it is impossible to accurately calculate the entire effective length of the antenna.

**Table 2.** Comparison of parameters at different effective lengths.

| Effective Length (μm) | | φ Direction | | | | θ Direction | | | |
|---|---|---|---|---|---|---|---|---|---|
| | | SLL | | η | | SLL | | η | |
| | | Smooth | Rough | Smooth | Rough | Smooth | Rough | Smooth | Rough |
| Si | 50 | −10.73 dB | −10.16 dB | 12.10% | 8.05% (−4.05%) | −12.87 dB | −12.56 dB | 70.15% | 67.23% (−2.92%) |
| | | | −11.03 dB | | 8.26% (−3.84%) | | −12.65 dB | | 67.78% (−2.37%) |
| | | | −9.63 dB | | 8.20% (−3.92%) | | −12.05 dB | | 67.41% (−2.74%) |
| | | | | Average: 8.17% (−3.93%) | | | | Average: 67.47% (−2.68%) | |
| | 70 | −10.83 dB | −10.85 dB | 13.45% | 8.98% (−4.46%) | −12.78 dB | −12.33 dB | 69.80% | 67.36% (−2.45%) |
| | | | −9.96 dB | | 8.48% (−4.97%) | | −12.25 dB | | 67.43% (−2.37%) |
| | | | −11.02 dB | | 8.60% (−4.85%) | | −12.45 dB | | 65.68% (−4.12%) |
| | | | | Average: 8.67% (−4.76%) | | | | Average: 66.82% (−2.98%) | |
| | 100 | −10.97 dB | −9.74 dB | 15.02% | 9.38% (−5.64%) | −12.81 dB | −12.00 dB | 70.37% | 66.76% (−3.61%) |
| | | | −10.88 dB | | 9.14% (−5.88%) | | −12.28 dB | | 66.95% (−3.42%) |
| | | | −9.68 dB | | 9.19% (−5.83%) | | −12.63 dB | | 66.19% (−4.18%) |
| | | | | Average: 9.24% (−5.78%) | | | | Average: 66.63% (−3.74%) | |
| | 130 | −11.15 dB | −10.51 dB | 14.89% | 9.73% (−5.16%) | −12.72 dB | −11.09 dB | 70.50% | 66.05% (−4.45%) |
| | | | −8.86 dB | | 9.24% (−5.65%) | | −12.25 dB | | 67.01% (−3.49%) |
| | | | −10.86 dB | | 9.77% (−5.12%) | | −11.99 dB | | 64.58% (−5.92%) |
| | | | | Average: 9.58% (−5.31%) | | | | Average: 65.88% (−4.62%) | |

## 3. Discussion

In addition to the rough surface simulated previously, there are other main types of fabrication errors that can negatively affect the far-field performance of antenna arrays. Therefore, in addition to the rough burrs created by the etching process on the surface, we further conducted random errors in the width along the antenna cells of ±5 nm on each antenna (@model 1) and random errors in the thickness of ±1 nm on each channel (@model 2). The width random error occurs when the standard deviation of the Gaussian distribution deviates from 0, indicating an overall deviation in the depth of the etching process. The thickness random error is caused by the potential random error introduced during the chemical polishing of the antenna arrays. As the entire wafer is significantly larger than the antenna array areas, we believe that the thickness error is relatively minor compared with the width error. The results presented in Figure 5 demonstrate that Si waveguides with random errors in width exhibit additional phase noise concentrated at certain random values. This leads to further chaotic interference, deterioration in the quality of far-field light spots in the φ direction and an overall energy tilt to one side, indicating the accumulation of significant phase noise due to the standard deviation. The SLL of only −4.55 dB indicates that it is more than four times higher than that of the ideal smooth antenna. The 7.66% decrease in $\eta$, compared with that observed for the smooth model mentioned earlier, indicates a limited detection distance. Additionally, the steering angle in the φ direction deviated by approximately 1.35° (as shown by the red line in Figure 5a). For the Si₃N₄ antennas, the impact of random errors in the width was almost imperceptible, with the same SLL and a familiar $\eta$ of a 0.005% difference from the rough surface model in the φ direction.

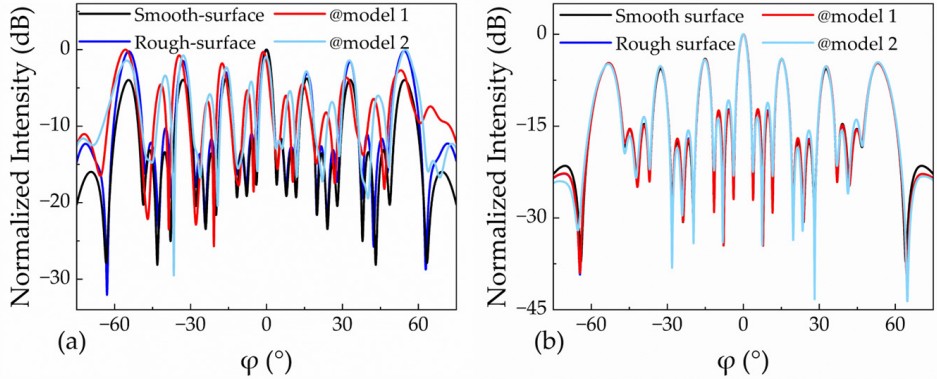

**Figure 5.** Far-field energy distribution in φ direction after introducing errors in the width and the thickness into antennas: (**a**) Si antenna; (**b**) Si₃N₄ antenna.

When an error in the thickness was introduced, a noticeable deterioration of the SLL and $\eta$ was observed in the energy distribution in the φ direction, disrupting the symmetry. Although the simulated error was slight, the resulting difference in widths after statistical calculation was significant. The SLL was only −3.79 dB, which means that the power of the sidelobe was almost half that of the main peak. The $\eta$ was only 7.16%, which is less than half of that of the smooth model. And there was a slight change in the θ direction as the $\eta$ changed to 65.69% compared with 66.63% in the rough surface model. However, the tilt was not as significant as the error in the width. This supports our previous conclusion that the standard deviation is concentrated around a certain value, causing the accumulation of phase noise. Additionally, there was no significant deviation in the steering angle. The random error in the thickness also had a greater impact on Si₃N₄ antenna arrays. The SLL was −11.05 dB and the $\eta$ was 17.95%, which is 0.60% lower than that of the smooth model. This indicates that the error in waveguide thickness was more significant than the error in waveguide width. The trend of the overall simulation results aligns with the actual situation shown in Figure 1.

To explore the deterioration of phase noise brought by fabrication errors and to compare the different performances between the two materials, we took the rough surface model and only used and simulated basic sidewall-etched antenna structures with limited channel numbers because of limited computing capacity. Obvious differences were observed, and conclusions and reasons are given. According to the results, the $Si_3N_4$ material has better tolerance in the process because of its smaller index and larger width. In the future research and design of OPA, a rough surface can be introduced to complex and functional designed structures, such as shallow-etched antennas or the $Si_3N_4$ perturbation antennas on Si waveguides that are mentioned in the Introduction. These results provide guidance on the fabrication stability of these structures.

## 4. Conclusions

Based on the actual etching process, this article introduces Gaussian distribution rough surface models based on Si and $Si_3N_4$ fishbone antennas. According to the simulation results, fabrication errors will affect the energy distribution of the Si antenna in the far-field and thus the quality of light spots, the SLL and bottom noise. When comparing $Si_3N_4$ with Si, it is evident that the deterioration of the $Si_3N_4$ antenna's far-field performance under the influence of fabrication errors is much slighter than that of the Si antennas, which aligns with the actual experimental results. In addition, when considering the fabrication error of waveguide width and thickness between antenna cells, $Si_3N_4$ antennas are also far superior to Si antennas. These findings suggest that $Si_3N_4$ antennas have better fabrication prospects.

**Author Contributions:** Conceptualization, P.W. and L.Y.; methodology, Y.X.; formal analysis, Z.W. and Y.Y.; investigation, P.M.; resources, W.L.; data curation, Y.X.; writing—original draft preparation, Y.X.; writing—review and editing, P.W. and Y.X.; visualization, Y.X. and L.C.; supervision, P.W. and J.P.; project administration, Y.Z. and J.P.; funding acquisition, P.W. and J.P. All authors have read and agreed to the published version of the manuscript.

**Funding:** This work was funded in part by the National Key R&D Program of China (2022YFB2804503), in part by the National Natural Science Foundation of China (62090053, 61934007 and 62105324), in part by the Opening Project of Science and Technology on Reliability Physics and Application Technology of Electronic Component Laboratory (21D04) and in part by the Strategic Priority Research Program of the Chinese Academy of Sciences (XDB43020202).

**Institutional Review Board Statement:** Not applicable.

**Informed Consent Statement:** Not applicable.

**Data Availability Statement:** Data are contained within the article.

**Conflicts of Interest:** The authors declare no conflicts of interest.

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
