# Peer review of "Analysis of the Influence of Fabrication Errors on the Far-Field Performance of Si and Si3N4 Antennas"

_photonics, doi:10.3390/photonics11010081_

Round 1

Reviewer 1 Report

Comments and Suggestions for Authors

The author simulated the performance of Si and SiN OPA with fabrication imperfections and concluded the advantage of SiN over Si. Below are my comments:

1.       Ambiguous expression in Abstract. What’s φ and θ direction respectively? Phase controlled and wavelength controlled? What do you mean by “While Si3N4 antenna is slightly impacted, about 1/9 of Si antenna in φ direction and 1/3 in θ direction (totally 1/27) “? What are you comparing? Can the deteriorations be multiplied?

2.       In figure 1, what are the designed parameters for the Si OPA and SiN OPA, respectively? It’s hard to reach a conclusion without the designed details.

3.       First of all, SLL should be a negative value instead of positive. How do you define SLL in θ direction? The side lobe is much higher than ~-30dB in blue line of figure 3(a) and figure 4(a), if you zoom in around the main lobe.

4.       Why is the smooth antenna has a SLL of -10.97dB, which is higher than theoretical -13.3dB? (page 4, line 100)

5.       Is it fair to compare Si waveguide of 500nm width and SiN waveguide of 1.5um width? You can actually design Si waveguide antenna of 6.5um width in reference: Dostart, Nathan, et al. "Serpentine optical phased arrays for scalable integrated photonic lidar beam steering." Optica 7.6 (2020): 726-733. It looks that multimode is not a problem in these wide waveguides. What will be the results if the width of Si and SiN are the same?

6.       This paper uses a model with random input to simulate the roughness, but only one result is shown in Figure 3 and 4 for rough antennas. It’s easy to choose the worst results from multiple random input to support your conclusion. A statistical simulation result should be shown here instead.  

7.       The model only simulates the standard deviation of 10nm. What about other values? More values are needed in your simulation.

8.       It’s questionable that the grating pitch will vary in fabrication. Even it’s varying, it will not vary randomly. I think you should consider simulating the height variations instead of pitch.

9.   What’s your point in simulating the effective length of antennas. What’s your conclusion? If you want to compare Si and SiN, why not simulate Si and SiN antenna with the same length. Again, one simulation is not enough, statistical simulation results are needed.

10.     Somehow experiment results are needed to support your conclusion. For example, fabricate a Si OPA and a SiN OPA with controlled parameters (same width, same lithography process and similar sidewall roughness).

Comments on the Quality of English Language

Gramma should be checked (for example, important significance, stabler…); figure numbers should be checked in main text (page 4 line 103).

Reviewer 2 Report

Comments and Suggestions for Authors

      The paper quantitatively studies the influence of fabrication errors on the performance of Si and Si3N4 antennas, and simulates them from three aspects: rough surface, random errors in widths and pitches. The structure and content of the article are clear, the simulation methods are reasonable, and the results have important significance, which prove that the influence of process errors on Si3N4 antennas is much smaller than that of Si (1/27 in total when only considering rough surfaces; and pretty remarkable with consideration of random width and pitch errors). Here are some suggestions and areas for improvement:

1. In this simulation, what is the setting of sources? And to simulate such precise structure correctly, what is the setting of meshes?

2. In actual fabrication, the random error in thickness of waveguide is also a kind of fabrication error. Why is the thickness error ignored here?

3. When discussing the influence with effective length, do the trends fit any functional relationship?

4. In terms of language, I suggest the expression “fabrication errors” in line 66 be changed to “rough surface”, and add structural phrase like “first of all”, as part 2 is mainly discussed about the influence of rough surface and other fabrication errors are discussed in part 3.

I suggest this manuscript for publication in Photonics after answer the questions.

Reviewer 3 Report

Comments and Suggestions for Authors

The manuscript provides an interesting analysis of the influence of the roughness determined by the fabrication process in the Far-Field performance of Si and Si3N4 antenna arrays. Results are obtained through FDTD simulations and show important deteriorations of the performance for Si antenna arrays. Si3N4 technology, on the contrary, is proven to be more robust to fabrication errors.

The paper is interesting and is potentially of great interest for researchers working in this field. However, in my opinion, should be revised to increase the overall quality and reach the standards required for the publication.

In particular:

1)     English should be improved. There are some sentences not terminated or introduced incorrectly in the context of the text (line 13-14, line 19-20, line 49-50, line 62-63, line 101-102, etc …);

2)     A figure with a clear description of the geometry should be introduced. Which are the effective dimensions of Si and Si3N4 waveguides and antennas?

3)     Do the authors have an explanation of the fact that Si and Si4N3 antennas perform differently with respect to the roughness determined by the fabrication process?

4)     Is the roughness model implemented for the two technologies (Si and Si3N4) realistic for the two fabrication processes?

Moreover, I think that the authors should better specify:

1)     The definition of the effective length of the antennas;

2)     The approach used to introduce a realistic model of the roughness in FDTD simulations. I’m expecting that simply reducing the pitch of the mesh to take into account the roughness end-up on an unpractical increase of the simulation time and of the memory requirements;

In the manuscript, the authors refer to some antennas ‘fabricated before’ (line 71): any reference for these fabrications? Moreover, on line 170 the authors refer to ‘well matching of actual experimental results’, but I do not see any experimental results used to confirm the simulations.

Comments on the Quality of English Language

English should be improved. There are some sentences not terminated or introduced incorrectly in the context of the text 

Round 2

Reviewer 1 Report

Comments and Suggestions for Authors

The author revised the paper from last version, but it’s still not qualified to be published. Please see my comments below. There are potentially more problems than stated here. I suggest you discuss the paper among all authors thoroughly especially senior members and resubmit after careful revision.

1.       In Abstract, “including rough surface, random error in width and pitches”. Pitch error is not simulated in main text, but remained in abstract. After this sentence, you claimed improvement in main peak power and SLL, but under what condition? Is the improvement only related to error in width, or both error in width and thickness? Obviously, it’s only related to error in width, but you mentioned the random error in width and thickness before your conclusion and leads to misunderstanding. Please rephrase your abstract and think about the logic.

2.       All data and conclusion in abstract should be explicitly state in main text. In Abstract, you conclude that Si deteriorates by 2.65dB in SLL in θ direction, but I can’t find it in main text.  The number 1.23 dB, 2.65 dB seems to come from one simulation, you need more simulations. 5.78% and 3.74% are only stated in table 1 and can’t be found in main text.  You conclude that SiN shows improvement by 1/15 in φ direction and 1/5 in θ direction, why are these numbers improved so much from last version? This is also the reason I suggest you do more simulations and have statistical results, two more simulations are not enough. You conclusion depends a lot on simulation times because it’s based on random initial input.

3.       It seems that your previous simulation data in table 1 (simulation for SiN) is inconsistent with the new data. Why do you abandon the data from previous simulation? Because it’s worse?

4.       Again, if you want to do fair comparison, you need to compare the Si and SiN antennas with the same effective length. Simulating SiN Antennas with longer effective length is a trick to have better results.

5.       In Discussion, the caption of figure 5 “errors in width and pitch”. Should it be pitch or thickness?

6.       I do not quite agree with your argument for single mode waveguide. You can find many OPAs which doesn’t use single mode waveguide. In the reference paper I listed, the FOV along the grating direction is not small, which is 35.8°. And for your given far-field image of multimode grating, it’s questionable that the multimode is stimulated in grating section or before grating section. I think you should limit your claim in abstract and conclusion to fish-bone like single mode grating antennas, and give your argument for single mode in main text.

7.       You need to analysis the root cause why SiN has better performance than Si, not just give you conclusion based on simulation. I think the main reason is simple which is that single mode SiN has wider waveguide and the optical field is less affected by the dimension error induced by fabrication process.

8.       “For OPA, although the field of view is limited in Si3N4 antenna because of large pitch between adjacent antennas, it can be greatly improved by using non-uniform arrays [28] or structures that designed separately for transmission and reception[40].” How is this paragraph related to the theme of your paper?

Comments on the Quality of English Language

1.       Please do proofreading before submission, there are still plenty of gramma errors; I can’t list them all here. Just name a few, in line 160, the noise is significantly increased should remove is; in line 243, mainly should be main; in line 265, an obvious deterioration of SLL and η be; in line 284, only use and simulate in basic should remove in

Round 3

Reviewer 1 Report

Comments and Suggestions for Authors

Below are my comments.

1.       As my comments in second round, all data and conclusion in abstract should be explicitly state in main text. However, I still can’t find where -0.51 dB comes from.

2.       As my comments in first round, What do you mean by “While Si3N4 antenna is slightly impacted, about 1/9 of Si antenna in φ direction and 1/3 in θ direction (totally 1/27) “? What are you comparing? Can the deteriorations be multiplied?  How can you multiply 1/9 and 1/3 and get 1/27 (it’s 1/15, 1/5 and 1/75 in the revised version) for the main peak power?

3.    As your response to my comment 1 in second round: The improvement related to both error in width and thickness. However, these numbers -0.87 dB, 5.78%, 3.74% all appears in main text in SECTION 2, which is about the rough surface model, which is only related to the roughness of width. These number all shows before you analysis the effects of thickness variation. Apparently, these numbers only relates to the error in width. But you mentioned thickness before these numbers and leads to misunderstanding that thickness also contributes to improvement of these numbers.

4.    Your response to my comment 8 in second round:

For OPA, although the field of view is limited in Si3N4 antenna because of large pitch between adjacent antennas, it can be greatly improved by using non-uniform arrays [28] or structures that designed separately for transmission and reception[40].” How is this paragraph related to the theme of your paper?

For OPA, the field of view of the Si3N4 array is more limited due to the larger spacing required between adjacent antennas, which is also a consequence of the smaller refractive index. However, this drawback can be greatly improved by using non-uniform arrays[28] or structures that are designed separately for transmission and reception[40].”

Yes, Non-uniform array will have larger FOV. However, Non-uniform array will significantly deteriorate the main peak power and SLL! It solves one problem but induces two more problems. Especially these two problems are your main claim of your paper!

Comments on the Quality of English Language

None
